# Isolation, Purification, and Structural Characterization of Polysaccharides from *Codonopsis pilosula* and Their Anti-Tumor Bioactivity by Immunomodulation

**DOI:** 10.3390/ph16060895

**Published:** 2023-06-19

**Authors:** Nan Li, Ying-Xia Xiong, Fan Ye, Bing Jin, Jin-Jia Wu, Miao-Miao Han, Tian Liu, Yi-Kai Fan, Cun-Yu Li, Jiu-Shi Liu, Ying-Hua Zhang, Gui-Bo Sun, Yun Zhang, Zheng-Qi Dong

**Affiliations:** 1Drug Delivery Research Center, Institute of Medicinal Plant Development, Peking Union Medical College & Chinese Academy of Medical Sciences, Beijing 100193, Chinaspring2378@163.com (F.Y.); bingjin021@163.com (B.J.); wujinjia212@163.com (J.-J.W.); hmmcgsa@163.com (M.-M.H.); hnlt0240@163.com (T.L.); fyk19981217@163.com (Y.-K.F.); jsliu@implad.ac.cn (J.-S.L.);; 2Key Laboratory of New Drug Discovery Based on Classic Chinese Medicine Prescription, Beijing 100700, China; 3Key Laboratory of Bioactive Substances and Resources Utilization of Chinese Herbal Medicine from Ministry of Education, Chinese Academy of Medical Sciences, Peking Union Medical College, Beijing 100094, China; 4Beijing Key Laboratory of Innovative Drug Discovery of Traditional Chinese Medicine (Natural Medicine) and Translational Medicine, Beijing 100700, China; 5Department of Pharmaceutics, School of Pharmaceutical Sciences, Heilongjiang University of Chinese Medicine, Harbin 150036, China; 6Department of Chinese Medicine Pharmaceutics, School of Pharmaceutical Sciences, Nanjing University of Chinese Medicine, Nanjing 210023, China; 7Jilin Academy of Chinese Medicine Sciences, Changchun 130012, China; zhangyinghua0214@126.com; 8Joint Research Center for Chinese Medicinal Herbs, IMPLAD, ABRC & ACCL, Beijing 100193, China

**Keywords:** *Codonopsis pilosula*, graded polysaccharides, macrophage polarization, antitumor activity, cascade membrane technology

## Abstract

The activity of polysaccharides is usually related to molecular weight. The molecular weight of polysaccharides is critical to their immunological effect in cancer therapy. Herein, the *Codonopsis* polysaccharides of different molecular weights were isolated using ultrafiltration membranes of 60- and 100-wDa molecular weight cut-off to determine the relationship between molecular weight and antitumor activities. First, three water-soluble polysaccharides CPPS-I (<60 wDa), CPPS-II (60–100 wDa), and CPPS-III (>100 wDa) from *Codonopsis* were isolated and purified using a combination of macroporous adsorption resin chromatography and ultrafiltration. Their structural characteristics were determined through chemical derivatization, GPC, HPLC, FT–IR, and NMR techniques. In vitro experiments indicated that all *Codonopsis* polysaccharides exhibited significant antitumor activities, with the tumor inhibition rate in the following order: CPPS-II > CPPS-I > CPPS-III. The treatment of CPPS-II exhibited the highest inhibition rate at a high concentration among all groups, which was almost as efficient as that of the DOX·HCL (10 μg/mL) group at 125 μg/mL concentration. Notably, CPPS-II demonstrated the ability to enhance NO secretion and the antitumor ability of macrophages relative to the other two groups of polysaccharides. Finally, in vivo experiments revealed that CPPS-II increased the M1/M2 ratio in immune system regulation and that the tumor inhibition effect of CPPS-II + DOX was superior to that of DOX monotherapy, implying that CPPS-II + DOX played a synergistic role in regulating the immune system function and the direct tumor-killing ability of DOX. Therefore, CPPS-II is expected to be applied as an effective cancer treatment or adjuvant therapy.

## 1. Introduction

The treatment of cancer mainly includes surgery, chemotherapy, radiation therapy, and immunotherapy, and the choice of therapy has undergone evolutionary changes as the understanding of the underlying biological process has improved. Among these, cancer immunotherapy allows disease treatment via artificial stimulation of the immune system. However, despite these advancements, there are still some limitations in the field of immunotherapy [1], which include the unpredictable treatment efficacy and patient response, the need for additional biomarkers, and the adverse immunological responses, especially the cytokine storm toxicity [2]. Therefore, the search for high-efficacy and fewer-toxicity immunomodulators has become the focus of current research.

*Codonopsis pilosula* (Franch.) Nannf., which belongs to the family Campanulaceae, has been used in food and prescribed in traditional folk medicine across Asian countries for thousands of years [3,4]. The roots of *C. pilosula* have long been prescribed for nourishing the spleen and tonifying Qi of the stomach [5,6]; importantly, *C. pilosula* presents significant immunomodulatory activity with a high safety [7,8]. Various types of compounds have been extracted from the roots of *C. pilosula*, including polysaccharides [9], lignans [10], flavonoids [11], sesquiterpenes [12], triterpenoids [13], and alkaloids [14]. Polysaccharides are one of the major biologically active substances responsible for their therapeutic functions. In recent years, several studies have reported that *C. pilosula* polysaccharides (CPPS) can exert antitumor effects by activating T lymphocytes, dendritic cells, and macrophages both in vitro and in vivo [4].

The activation of macrophages is necessary for immune system stimulation. Macrophages are an important part of the innate immune response and play a vital role in inhibiting the occurrence and metastasis of tumors [15]. In addition, macrophages display plasticity and can differentiate into different types under the influence of environments. Several polysaccharides are known to affect the immunomodulatory activity of macrophages [16], and they can induce tumor cell apoptosis through proinflammatory cytokines released by macrophages. Subsequently, the function of polysaccharides derived from *C. pilosula* to activate and repolarize macrophages has gained attention in the past years. Several studies have suggested that CPPS can accelerate the generation of NO, TNF-α, and IL-6, as well as increase ERK1/2 and JNK by activating the MAPK signaling pathway [17]. For instance, a study revealed that the crude polysaccharides from *C. pilosula* could inhibit the proliferation of IL-4-induced M2-like TAMs and significantly increase the mRNA expression levels of IL-1, IL-6, iNOS, and TNF-α, thereby reducing the tumor volume in melanoma mice via repolarization of tumor macrophages [18]. However, the crude CPPS also contains several neutral sub-polysaccharides, and each section may exhibit different characteristics, including their molecular weights, specific glycosidic linkages, monosaccharide compositions, the degree of branching, polymerization, and the spatial configuration of the chains [19]. In addition, the relationship between the detailed structures of each CPPS and the pharmacological activities of tumor cells remains unclear, warranting further research.

In the current study, we attempted to grade CPPS based on the average molecular weights of polysaccharides by using the membrane-separation method. The membrane-separation technology is a moderate and feasible approach to classify polysaccharides with different molecular weights [20]. Ultrafiltration membranes have been effectively used to grade polysaccharides extracted from camellia seed cake and corn in other studies [21,22]. The structural characteristics of different polysaccharide fractions were investigated by NMR spectroscopy, gel permeation chromatography, Sephadex G100 gel (GE, Uppsala, Sweden) permeation, and monosaccharide composition analysis, and the immune regulation activity in RAW264.7 macrophages of different purified polysaccharides was analyzed by in vitro experiments. Furthermore, the potential effect of polysaccharides on melanoma in C57 mice was explored, and the impact on tumor-associated macrophages was investigated. This is the first study to demonstrate the potential relationship between graded CPPS structure and TAM immunomodulatory activity and examine the possible molecular action mechanisms. Our results demonstrated that moderate molecular weight is the most critical immunomodulator in the anti-melanoma effect and that it may be helpful to develop specific polysaccharide-based pharmaceutical preparations.

## 2. Results

### 2.1. Purification and Structural Characterization of Polysaccharides

The *Codonopsis* polysaccharides were first prepared via water extraction and alcoholic precipitation, and a certain amount of refined polysaccharides (85.6% purity) was obtained by the Sevage method to remove the protein components (Figure 1A). We obtained three polysaccharide fractions of *Codonopsis* polysaccharides with different molecular weights separated by two different pore-size membranes; these three polysaccharides are deemed the most important components of the crude polysaccharide of *Radix Codonopsis*. After the three polysaccharides were obtained at relative abundances of 28.8%, 8%, and 63.2%, respectively, the purity of the polysaccharides was measured by Sephadex G-100 chromatography (Figure 1B), and CPPS-I, CPPS-II, and CPPS-III exhibited single symmetrical peaks in both the G-100 chromatogram, indicating that both were homogeneous polysaccharides of high purity.

#### 2.1.1. Chemical Composition Analysis

The monosaccharide composition analysis (Figure 1C) with reverse-phase HPLC according to PMP derivatization procedures indicated that CPPS-I and CPPS-II were mainly composed of glucose, mannose, and galactose, and glucose was found to be the major monosaccharide in these fractions, indicating that CPPS-I and CPPS-II are two glucose-rich heteropolysaccharides. CPPS-III was mainly composed of mannose, galacturonic acid, and glucose at a molar ratio of 2.30:1.62:1, respectively. However, Liu et al. reported that a *C. pilosula* neutral polysaccharide (CPEP) consisted of glucose and galactose [23]. Zou et al. reported an acidic polysaccharide from *C. pilosula* Nannf. Var. modesta (Nannf.) L.T. Shen (WCP-I) was composed of a rhamnogalacturonan region containing arabinogalactan as sidechains [24]. The difference in the monosaccharide composition can mainly be attributed to the difference in species, cultivation regions, and extraction methods.

The homogeneity of CPPS-I, CPPS-II, and CPPS-III was identified by HPGPC-RID as a single peak with the polydispersity index (PDI) being 1.56, 1.35, and 1.42, respectively (Figure 1D). The average molecular weights (Mw) of different CPPS were determined to be 5.7 × 10^5^, 9.8 × 10^5^, and 3.3 × 10^6^ Da for CPPS-I, CPPS-II, and CPPS-III.

#### 2.1.2. The Rheological Analysis

Since the polysaccharides are rich in hydroxyl groups, their gelation behavior may have an impact on their subsequent pharmacological processes. Therefore, the viscosity of the polysaccharides (Figure 1E) was measured, and the viscosity of the polysaccharide solutions of CPPS-I, CPPS-II, and CPPS-III was found to demonstrate a rapid decrease in the shear rate range of 0.1–10 s^−1^ with a gradual increase in the shear rate, and the solutions were shear thinning. When the shear rate exceeded 10 s^−1^, the viscosity of the solution gradually stabilized and demonstrated the typical “pseudoplastic fluid” type, which was due to the shear force straightening the internal curly connected molecular structure of the solution and reducing the entanglement points. The loss angle tangent (tanδ) is the ratio of G″ to G′, and tanδ < 1 indicates that the viscoelastic material mainly embodies elastic characteristics. The frequency scan results depicted that the energy storage modulus (G′) of polysaccharide increases uniformly with an increase in the oscillation frequency, and there are small fluctuations in the oscillation frequency range of 1–10 Hz, and G′ is greater than G″, indicating that the polysaccharide has gel properties at 0.1–10 Hz at a strain of 0.1%. The relaxation modulus results indicated that the relaxation modulus tends to stabilize and then increase with increasing polysaccharide mass fraction, thereby indicating that increasing polysaccharide concentration prompts increased cross-linking of intermolecular hydrogen bonds; therefore, the gel network structure is tighter and exhibits better rheological properties.

#### 2.1.3. UV, FT-IR, and NMR Analyses

The UV–vis spectra of CPPS-I, CPPS-II, and CPPS-III (Figure 2A) indicated no typical peaks at 260 or 280 nm, which proved no presence of protein or nucleic acid in CPPS samples. The results suggested that the polysaccharides were of high purity (Figure 2A). Figure 2B illustrates the IR spectra of the CPPS-I-III; these polysaccharides display a similar spectrum. A broad absorption of around 3400 cm^−1^ was attributed to hydroxyl stretching vibration, and the band in the region of 2932 cm^−1^ was produced by the C-H stretching vibration. A strong band at 1634 cm^−1^ was detected in association with water, and the characteristic bands around 1130 cm^−1^ and 1032 cm^−1^ were assigned to the pyranose ring C-O stretching vibration. Furthermore, the signal at 935 cm^−1^ was detected on behalf of the D-glucose pyranose ring asymmetrical stretching vibration. The absorption band at 870 cm^−1^ and 600 cm^−1^ indicated the formation vibration of α-formation C-H in the pyranose ring and pyranose ring skeleton, respectively. Therefore, it can be concluded that all CPPS-I-III belong to α-type glucan with the pyranose group.

The structure feature of CPPS-I-III was elucidated by 1DNMR spectroscopy [4]. Generally, the anomeric signals concentrated in the range of δ5.0–5.8 ppm in ^1^H-NMR and δ97–101 ppm in ^13^C-NMR represented the type of β-configuration. The δ4.4–5.0-ppm region and δ103–107 ppm represented the α-configuration. As shown in Figure 2C, the main anomeric proton signals of three polysaccharides at δ5.0–5.8 ppm are presented, indicating that sugar residues in CPPS-I-III mainly presented with α-configuration. The chemical shift from δ3.60 to 4.20 ppm in the ^1^H NMR was attributable to protons from C-2 to C-6 in the residue.

#### 2.1.4. Morphological Properties

After identifying the difference in primary structure, we observed the spatial morphology of CPPS-I-III by Atomic Force Microscopy (AFM) and Scanned Electron Microscopy (SEM). Firstly, the morphology of CPPS-I–III polysaccharides on mica sheets was scanned and imaged by AFM to visualize the nanoscale structure. As shown in Figure 2D, CPPS-I with larger molecular weight demonstrated a complete ring structure and a longer chain structure. In contrast, in CPPS-I and II, the molecular chains did not show a multi-branched structure or longer chain structure. Rather it showed only several polysaccharide aggregates of different sizes, creating a lamellar structure. SEM results showed that the polysaccharide surface of CPPS-I–III was displayed as a rough and irregular lamellar structure at 100× magnified times (Figure 2E). Comparatively, when CPPS-I and III were magnified 5000× times, the polysaccharide fragment surface was observed to be slightly uneven and with a few holes. However, CPPS-II showed a stick cluster-like stacked structure at 5000× magnification. The difference in the structure between CPPS-II and CPPS-I, III may induce different bioactivities.

### 2.2. In Vitro Experiments

#### Antitumor and Immunomodulatory Activity of Three Polysaccharides

CCK-8 assay was performed to analyze the impact of CPPS-I–III on RAW264.7 (Appendix A) and B16F10 cell viability. As shown in Figure 3A, cell proliferation was inhibited by CPPS-I–III treatment, while CPPS-I-III inhibited cell viability in a dose-dependent manner. In addition, the treatment of CPPS-II exhibited the highest inhibition rate at a concentration of 1000 μg/mL among all groups, which was almost as efficient as the DOX (10 μg/mL) positive-control group at the 125 μg/mL concentration. A past study showed that the polysaccharides in *C. pilosula* may induce tumor cell apoptosis by activating the mitochondrial apoptotic pathway. The mechanism of apoptosis may be related to increasing the mitochondrial membrane permeability [25], promoting the release of cytochrome C, and activating downstream caspase 3 through a cascade reaction.

To further investigate the impact of CPPS-I–III on macrophages, the release of NO in the supernatants was measured by the Griess reaction. Chemicals called “nitrogen radicals” produced by the upregulation of inducible NO synthase iNOS marked the increased pathogen-killing ability. As shown in Figure 3B, the secretion of NO was significantly enhanced after LPS stimulation for 24 h. The levels of NO began to increase by 1.25-folds compared with the control group when the CPPS-II concentration increased to 1000 µg/mL. In contrast, treatment with CPPS-I and III at different concentrations showed no effect on the basal level of NO in RAW264.7 cells.

Flow cytometry was used to examine the expression of CD86, an M1-type marker, on the surface of RAW264.7 macrophages after 24 h of stimulation with Three polysaccharides. As shown in Figure 3E, CD86 expression increased significantly compared to the control group, demonstrating a shift in the RAW264.7 cells from the initial M0 state to the M1 phenotype. Among the three groups of polysaccharides, CPPS-II had the strongest CD86 expression ability, indicating that CPPS-II had the stronger ability to promote M0 differentiation into the M1 type and had a stronger immunomodulatory ability.

Firstly, RAW264.7 was co-cultured with indirectly exposed B16F10 (melanoma) cells by using the transwell insertion method (Figure 3C) to study the effect of CPPS-II on macrophages and cancer cells. RAW264.7 cells were seeded on inserts made of a membrane of 0.4 μm pores, which allowed the exchange of soluble factors, but not the trans-migration of cells. As shown in Figure 3D, the apoptosis rate of tumor cells remained unchanged irrespective of whether macrophages existed, indicating no impact on tumor cells when the macrophage remained unstimulated. Despite the slightly inhibitory effect of CPPS-II on tumor cells, both CPPS-I–III did not cause tumor cell apoptosis in the absence of RAW264.7. However, treatment with CPPS-II dramatically increased the tumor cell apoptosis rates, while taxol positive group also enhanced the tumor cell apoptosis rate to confirm the reliability of this study. These results were consistent with the tumor cell proliferation and NO secretion results mentioned before. Although the CPPS-II treatment exhibited antitumor activity to a certain extent, the treatment showed significantly higher cell apoptosis rates in the co-cultured model. Therefore, it can be inferred that CPPS-II can enhance the percentage of apoptotic cells by activating macrophages. The potential mechanism may be related to promoting the secretion of inflammatory factors by macrophages.

To investigate the mechanism through which CPPS-II induces apoptosis in cancer cells, a cell cycle assay was performed in B16 cells. In the co-culture system of B16 and macrophages without CPPS-II, the proportion of cancer cells in the G1 phase increased slightly compared to the control group. As shown in Figure 3E, the proportion of cancer cells in the G1 stage increased from 37.04% to 47.41% and 73.10% after CPPS-II administration in the co-culture system. This finding indicates that CPPS-II exerts its inhibitory function mainly by inhibiting the differentiation of tumor cells from the G2/M phase to the G0/G1 phase.

### 2.3. In Vivo Experiments

#### 2.3.1. Antitumor Effect In Vivo

Encouraged by the effective antitumor performance in vitro of the CPPS-II + DOX treatment group, B16F10 xenograft mouse models were used to evaluate the antitumor effect of co-treatment therapy in vivo. As shown in Figure 4A–C, CPPS-II, DOX, and CPPS-II + DOX groups showed a significant reduction in the growth rate of the tumor. It was beneficial from the individual chemotherapy, TAM-repolarized immunotherapy, and the combined effect. The tumor growth inhibition level of the combined CPPS-II + DOX treatment was superior to that of DOX and CPPS-II treatments individually, indicating the advantage of multiple mechanisms acting simultaneously.

TUNEL and H&E staining assays were performed to further investigate the comprehensive antitumor activity of co-treatment therapy. As shown in Figure 4D, the co-treatment group exhibited the most severe cell apoptosis area when compared to other groups in TUNEL images, and obvious chromatin condensation and shrinkage of tissue structures were observed in the H&E staining photos.

Ki67 is a nuclear protein that could indicate cell proliferation activity; the Ki 67-positive tumor cells are often correlated with the clinical course of cancer. As shown in Figure 4E, the control group demonstrated a stronger positive expression. In contrast, all other groups showed a weaker yellow color after administration, indicating that the cell proliferation activity was weakened. Subsequently, the expression of other markers, CD31 and MMP9, was used to demonstrate the degree of tumor angiogenesis and metastasis, respectively. As shown in Figure 4E, the co-treatment group significantly downregulated the CD31 and MMP9 expression in the tumor tissues when compared to the DOX or CPPS-II treatment groups individually.

#### 2.3.2. TAMs Population and Cytokine Expression Profiles in the TME

The intratumoral infiltration of TAMs (CD45 + CD11b + F4/80+) was measured to reflect the impact of different treatment groups on the tumor immune environment. However, TAMs are heterogeneous immune cells that have been historically categorized into two groups, as follows: M1 (MHC-I) and M2 (CD206). M1 macrophages play an important role in proinflammatory and cytotoxic (anti-tumoral) functions, whereas M2-TAMs refer to anti-inflammatory and tissue healing (pro-tumoral) functions. Both the subtypes of TAMs were analyzed by flow cytometry and immunofluorescence staining. As shown in Figure 5A–D, the frequency of M2-TAMs after polysaccharide treatment alone and co-treatment were distinctly lower than that in the saline group and DOX group individually. Meanwhile, the number of M1-TAMs treated with CPPS-II and CPPS-II + DOX was the highest among all groups.

In addition, the secretion of immune cytokines was measured to reflect the local immune environment of the tumor. According to the different types of cytokines secreted by macrophages, they can be divided into the following subgroups: Th1 and Th2. Th1-like cells secrete cytokines such as tumor necrosis factor (TNF-α), while Th1-like cells secrete cytokines such as tumor growth factor IL-10; these two types of cells are involved in different immune responses. ELISA measures Th1 cytokine (TNF-α) levels and Th2 cytokine (IL-10). As shown in Figure 5E,F, the expression of Th1 cytokine was promoted while that of Th2 cytokine was decreased. These data also indicate that CPPS-II modified the tumor microenvironment and promoted antitumor therapy by repolarizing M2 TAMs to M1 TAMs [26].

Fluorescence microscopy observations confirmed a decrease in the expression of M2 (CD206 marker) in tissue macrophages after administration via CPPS-II (Figure 6A); the expression of iNOS was increased. This finding is consistent with the flow cytometry data, which suggests that CPPS-II can modify the tumor microenvironment by repolarizing M2 TAMs to M1 TAMs. The increased expression of iNOS produced nitric oxide (NO) to achieve the tumor-killing effect, indicating that CPPS-II has a good regulatory effect on TAMs [27].

#### 2.3.3. Signaling Pathway Determinations

To further investigate the underlying mechanism of co-treatment therapy, we examined several major downstream signaling molecules of tyrosine kinases, including Stat1 and Stat3, in tumor tissues isolated at the end of the pharmacodynamic assay. The results demonstrated that CPPS-II monotherapy significantly decreased the p-Stat3 and Stat3 protein levels compared to the saline and DOX groups (Figure 7A–C). Moreover, the expression of p-Stat3 and Stat3 after CPPS-II + DOX treatment remained unchanged relative to that after CPPS-II treatment alone. The abnormally high levels of Stat3 activity have been associated with an increased likelihood of melanoma returning after treatment. The activation of Stat3 promoted cell division and reduced cell apoptosis, and the inhibition of p-Stat3 inhibited tumor growth by inducing tumor cell apoptosis [28], increasing dendritic cell activation, reducing tumor Treg and CD8+ T-cell activation, and decreasing the accumulation of MDSCs. We also measured the p-Stat1 and Stat1 protein expression increase after treatment with CPPS-II and CPPS-II + DOX. The activation of Stat1 could enhance the ability of M1-polarized macrophages and elevate the production of proinflammatory cytokines [29] (Figure 7D,E). Therefore, together with the results discussed above, CPAP may drive M2-type macrophage polarization by promoting Stat1 phosphorylation and inhibiting Stat3 activation. The original western data were displayed in Appendix A.

### 2.4. Safety Evaluation

Toxicity and safety evaluation is an important aspect of novel therapy evaluations. The major organs (such as the hearts, liver, spleen, lung, and kidney) from different groups were harvested and stained with H&E. As shown in Figure 8A, the morphological images indicated that DOX treatment induced an injury to the nontargeted tissues, especially in the heart. No significant morphological changes were detected between the groups receiving CPPS treatment only and the combination treatment. According to a past study, DOX-induced cardiotoxicity can be attributed to the abnormal level of oxidative stress or inflammation. When compared with chemotherapy, CPPS-II can effectively avoid toxic outcomes according to the mechanism of re-educating tumor-associated macrophages and even attenuate DOX toxicity in the combined therapy. The other component of the toxicity study (Figure 8B), body weight loss, also showed no difference among the different groups. Together, the protective properties of CPPS-II against DOX toxicity suggest a promising combination therapy for clinical application.

## 3. Discussion

In this study, the crude polysaccharides of *Codonopsis pilosula* were extracted through water extraction and by using the alcohol precipitation method, and the proteins obtained in the crude polysaccharides were removed via the Sevage method to obtain the refined polysaccharides. In the ultrafiltration system (membrane-separation method), the polysaccharide fragments of different molecular weights (CPPS-I < 60 wDa, 60 wDa < CPPS-II < 100 wDa, and CPPS-III > 100 wDa) were obtained by using ultrafiltration membranes with molecular retention of 60 and 100 wDa, respectively, according to the molecular weight distribution. CPPS-I and CPPS-II are two glucose-rich heteropolysaccharides, mainly composed of glucose, mannose, and galactose. In contrast, CPPS-III mainly comprises mannose, galacturonic acid, and glucose. By UV, FT-IR, and NMR spectra, CPPS-I-III all belong to the α-type glucans containing pyranose moieties.

The molecular weight of polysaccharides isolated from the roots of *Codonopsis pilosula* varies, and different molecules of polysaccharides may have different biological activities [30]. Therefore, we investigated the antitumor activity and the immunomodulatory potential of different molecular weight polysaccharides from *Codonopsis pilosula*. The results revealed that all polysaccharides from *Codonopsis pilosula* demonstrated good antitumor activity, but only CPPS-II exhibited some immunomodulatory ability. From the study of the in vitro antitumor activity of *Codonopsis* polysaccharides, CPPS-I–III showed a good tumor inhibition rate against tumor cells. However, the antitumor effect of CPPS-II was much greater than that of CPPS-I and CPPS-III. The tumor inhibition rate of CPPS-II at 30 μg/mL concentration was comparable to that of the positive-control group DOX. Second, both CPPS-I-III and LPS showed significant inhibitory effects on macrophages, albeit the toxicity to macrophages could not be relied on to reflect the immune functions. Therefore, co-culture experiments were performed to investigate macrophages and tumor cells. In the co-culture experiments, all macrophages treated with polysaccharides increased the tumor suppression rate. Still, CPPS-II increased the tumor suppression rate most significantly, followed by CPPS-II-led increase of the NO expression level of macrophages and improved tumor-killing ability. At the same time, CPPS-I and CPPS-III showed no significant change in the NO expression level.

Adriamycin is a broad-spectrum antitumor drug [31] that can effectively treat mid- to late-stage tumors as well as tumors with a propensity to disseminate, albeit it induces damage to normal cells and leads to immune system destruction and organ toxicity, which limits its further application. Therefore, effectively overcoming the adverse effects of chemotherapeutic drugs could improve the efficiency of cancer treatment. *Codonopsis* polysaccharide, as an active component of traditional Chinese medicine, confers the effects of improving the immune function [32] and anti-aging [33], promoting metabolism [34], and improving body functions [35], which are beneficial to enhance the antitumor effect of chemotherapy drugs and improving the damage caused by chemotherapy drugs to patients. Therefore, we further explored the mechanism of CPPS to promote adriamycin tumor treatment, enhance immune function, and reduce its toxic side effects for antitumor treatment [36]. First, CPPS-II, with the best efficacy, was selected from CPPS-I–III by in vitro experiments as the study of in vivo experiments. In the in vivo antitumor experiments, CPPS-II + DOX displayed better tumor suppression than polysaccharides or DOX administration alone, which facilitated the strategy of chemotherapy combined with polysaccharide immunotherapy. To investigate the regulation of immunity by polysaccharides, the phenotype of tumor tissue macrophages was determined by flowmetry, and the results revealed that the number of M1-TAMs was highest in the CPPS-II-containing administration group. In contrast, the number of M2-TAMs was significantly lower than that in the saline and DOX alone treatment groups. Second, CPPS-II monotherapy significantly decreased the p-Stat3 and Stat3 protein levels and increased the p-Stat1 and Stat1 protein expression. Considering the abovementioned experimental results, CPPS-II is likely to drive repolarization of M2 TAMs to M1 TAMs by promoting Stat1 phosphorylation and inhibiting Stat3 activation to modify tumor-associated macrophages in order to achieve tumor treatment effect through immunotherapy. H&E staining of the tumor tissues in the control group demonstrated typical pathological features of tumors. TUNEL and Ki67 results also revealed that the CPPS-II + DOX treatment group produced the most pronounced inhibition of tumor cell proliferation, indicating that the combination therapy produced a good therapeutic effect. Subsequently, the expression of markers such as CD31 and MMP9 were used to indicate the extent of tumor angiogenesis and metastasis, respectively, and the CPPS-II + DOX group significantly downregulated the expression of CD31 and MMP9 in the tumor tissues and reduced the migration ability of tumors. CPPS-II alleviated the weight loss and normal tissue damage induced by DOX drugs in the animals. These results demonstrated that CPPS-II could enhance the antitumor of DOX, regulate tumor-associated macrophages, and alleviate the damage of chemotherapeutic drugs to the organism, indicating that CPPS-II + DOX had good prospects.

## 4. Materials and Methods

### 4.1. Materials and Chemicals

The roots of *Codonopsis pilosula* were collected from Weiyuan, in the Shanxi province of China, and were identified by Dr. Jiushi Liu (The Institute of Medicinal Plant Development, affiliated with the Chinese Academy of Medical Sciences and Peking Union Medical College). Sephadex G-100 was purchased from GE Healthcare Life Science (Uppsala, Sweden). Monosaccharide standards, DMSO, Lipopolysaccharide (LPS), and ethylenediamine were obtained from Sigma (St. Louis, MO, USA).

RPMI-1640 and fetal bovine serum were purchased from Gibco (Carlsbad, CA, USA). Cell counting kit-8 (CCK-8) and Annexin V-FITC Apoptosis Detection Kit were obtained from Beyotime (Shanghai, China). Assay kits for interleukin-6 (IL-6), interleukin-10 (IL-10) tumor necrosis factor-alpha (TNF-α), and BCA were all obtained from Nanjing Jiancheng Bioengineering Institute (Nanjing, China). All other chemicals and solvents used were of analytical reagent grade and obtained from Sinopharm Chemical Reagent Co., Ltd. (Ningbo, China).

### 4.2. Polysaccharide Extraction and Purification

*Codonopsis pilosula* polysaccharide was isolated from the dried body powder using hot-water extraction followed by ethanol extraction. The materials were extracted three times with boiling water (1:20, *w*/*v*, 2 h for each time), and the collected solution was evaporated to a certain volume to obtain the concentrate. An equal volume of ethyl acetate was added to the concentrate and mixed for at least 5 min. The top organic layer was carefully removed using a pipe and discarded. This step was repeated thrice, and the bottom aqueous was collected for further purification. Finally, the concentrate was precipitated by 95% ethanol (80% of eventual concentration) for 4 h at room temperature and collected the precipitate. The precipitate was redissolved in pure water and lyophilized for 48 h to yield *Codonopsis pilosula* crude polysaccharide(C-CPPS).

The Sevage method was used to remove the protein in the crude polysaccharide to yield *Codonopsis pilosula* refined polysaccharides (CPPS), the CPPS was intercepted according to the Mw distribution by using ultrafiltration membrane with molecular retention of 60 and 100 wDa in the ultrafiltration system. Each fraction was collected and lyophilized to give a white powder designated CPPS-I (<60 wDa), CPPS-II (60–100 wDa), and CPPS-III (>100 wDa). The purity of the preparation was measured by a Sephadax G-100 column (2.6 cm × 100 cm, GE Tech., Uppsala, Sweden) equilibrated with distilled water. Each polysaccharide was prepared for further immunological analysis.

### 4.3. Polysaccharide Characterization

#### 4.3.1. Molecular Weight and Homogeneity

Weight average molecular weight (Mw) of CPPS and sub-fractions were performed according to the report described by Li [37]. PEO standards were used to establish a calibration curve. The analysis was conducted on a gel permeation chromatography system (Shimadzu LC20, Shimadzu Co., Kyoto, Japan) equipped with a refractive index detector (Shimadzu RID-20, Shimadzu Co., Kyoto, Japan). The samples were loaded into a TSK Ultrahydrogel™ linear column (8 × 300 mm) coupled with a TSK Ultrahydrogel™ guard column (6 × 200 mm), and the column temperature was set at 35 °C. The 0.1 N nano3 solution containing 0.06% (*w*/*v*) nan3 was used as the mobile phase at the flow rate of 0.6 mL/min and the injection volume was 20 μL. Data was collected by Shimadzu labsolutions HW200 workstation.

#### 4.3.2. Rheological Properties of CPPS

Three kinds of CPPS-I, CPPS-II and CPPS-III polysaccharides were prepared, and their apparent viscosity was determined by Kinexus Lab + rheometer at 25 °C, the storage modulus G′ and loss modulus G′ of three kinds of *Codonopsis pilosula* polysaccharides solutions were determined by frequency scanning at 25 °C under 1% strain. The relaxation modulus g(t) PA of CPPS in 1%, 2%, and 3% solutions were calculated at 25 °C.

#### 4.3.3. Qualitative Analysis of Monosaccharide Composition

The monosaccharide composition of the polysaccharide was analyzed by reverse-phase HPLC according to PMP derivatization procedures [38]. Briefly, the standard monosaccharides were dissolved in deionized water and saved for PMP-label. The polysaccharides solution (2 mg/mL) and trifluoroacetic acid (4 M) were mixed for at least 5 s, then the mixture was sealed and incubated at 110 °C for 4 h in a heating block. Then, the reaction mixture was evaporated to dryness to remove residual TFA. The residue samples were reconstituted in pure water, and the standard samples were prepared as described above. Then, solution samples were dissolved in 0.6 M sodium hydroxide (50 uL), and 0.5 M PMP (100 uL), and the mixture was incubated at 70 °C for 1 h in a heating block. The mixture was neutralized by 0.3 M HCl solution and filtered through a 0.22 μm membrane. Analysis of the resulting solution was carried out using Agilent Technologies 1260 series apparatus equipped with DAD detectors and an Agilent Eclipse XDB C18 (150 mm × 4.6 mm). the flow rate was 1 mL/min, and the UV absorbance of the effluent was monitored at 250 nm. Mobile phases A and B were 0.1 M ammonium acetate, pH5.5, containing 15% acetonitrile, respectively.

#### 4.3.4. UV and IR Spectrum Analysis

The UV-vis absorption spectra of polysaccharide samples from 200 nm to 400 nm were recorded with a UV spectrophotometer (Agilent, Santa Clara, CA, USA). The FT-IR spectrum for the polysaccharide in the 4000–400 cm^−1^ wavelength was obtained with a Bruker vertex spectrometer (Bruker, Karlsruhe, Germany) at room temperature.

#### 4.3.5. NMR Analysis

The hydroxyl protons of polysaccharides were exchanged with D2O three times. Samples were lyophilized and dissolved in D2O at a 20–30 g/L concentration. all NMR spectra were recorded using an NMR spectrometer (Jeol, Kyoto, Japan).

#### 4.3.6. SEM Analysis

Scanning electron microscopy (SEM), images were observed with field emission scanning electron microscopy (Jeol, Kyoto, Japan). The samples were sputtered with gold and placed onto double-sided adhesive tape attached to a circular specimen stub for analysis.

#### 4.3.7. AFM Analysis

The molecular morphology of polysaccharides was recorded using a ScanAsyst AFM (Bruker, Karlsruhe, Germany).

### 4.4. In Vitro Evaluation

#### 4.4.1. Cell Culture

RAW264.7 and B16 cells were purchased from the Cell Resource Center of the Chinese Academy of Medical Sciences & Peking Union Medical College (Beijing, China). Both were maintained in DMEM high-glucose medium supplemented with 100 μg/mL streptomycin, 100 U/mL penicillin, and 10% heat-inactivated fetal bovine serum. The cells were cultured at 37 °C in a humidified atmosphere containing a 5% CO_2_ incubator.

#### 4.4.2. Cell Viability Assay

The effect of the polysaccharides on the viability of RAW264.7 and B16 cells was determined by the CCK-8 method. Briefly, RAW264.7 and B16 cells were seeded 100 μL at a density of 1 × 10^5^ cells/mL in 96-well plates, respectively, and cultured overnight. CPPS-I, CPPS-II, and CPPS-III were added at final concentrations of 0, 62.5, 125, 250, 500, and 1000 μg/mL to B16. After 24 h of incubation at 37 °C in a humidified, 20 μL of the cck-8 reagent were added to each well. After 1 h of incubation, the absorbance was recorded at 560 nm by using an ELISA plate reader and then translated into macrophage viability ratio for comparison. All determinations were performed for three replicates for every sample, and three independent assays were conducted.

#### 4.4.3. Measurement of NO

Logarithmically grown RAW264.7 cells were inoculated in 96-well culture plates at 1.0 × 10^4^ cells per well, incubated for 24 h at 37 °C with 5% CO_2_ saturation, and then treated with CPPS-I, CPPS-II, and CPPS-III (0, 62.5, 125, 250, 500 and 1000 μg/mL) or LPS (1 μg/mL), followed by incubation for another 12 h. The supernatant was collected from each well for measurement of NO. The NO level was measured according to manufacturer’s instructions of NO detection kit (Beyotime Biotechnology Institute, Shanghai, China).

#### 4.4.4. Macrophage Polarization

RAW 264.7 cells were inoculated in 6-well culture plates at 1.0 × 10^5^ cells per well and treated with three CPPS (50 μg/mL) for 24 h. At the end of the culture, the cells were collected in centrifuge tubes, washed three times with PBS, and incubated with APC-CD11c antibody for 30 min at 4 °C in the dark. Then, the expression levels of surface receptors were determined by flow cytometry (LE-SH800SA).

#### 4.4.5. Cell Cycle

Cell cycle was performed as previously described [39,40]. Briefly, B16 cells were seeded in 6-well plates at 1.0 × 10^6^ cells per well and mediated by various concentrations of CPPSs. After 48 h of incubation, cells were routinely digested with trypsin and collected followed by wash with cold PBS. Then, cells were fixed with 70% ethanol at 4 °C overnight and treated with 100 mg/mL RNasesolution and PI staining at 37 °C for 30 min. The samples were detected at the wavelength of 488 nm using a flow cytometer (FACS Canto, BD Biosciences, Franklin Lakes, NJ, USA), and the relative proportions of B16 cells in each phase were analyzed with Modfit software (Verity Software House Inc., Topsham, ME, USA).

#### 4.4.6. Annexin V-FITC/PI Detection

The extent of cell apoptosis was assayed using Annexin V FITC PI apoptosis kit. B16 cells (1 × 10^6^ cells/mL) were seeded in 6-well plates and treated with three CPPS (50 μg/mL) for 48 h. Cells were collected following digestion with EDTA-free trypsin, wash with PBS and centrifugation for 5 min at 1000 rpm. Subsequently, cells were subjected to 5 μL of Annexin V-FITC and 5 μL of PI were added and incubated at room temperature in the dark for 10 min. Cells were then filtered through a 40 μm cell filter and tested by flow cytometry.

### 4.5. In Vivo Anti-Tumor Activity Evalution

#### 4.5.1. Tumor Xenograft Models

Sub-confluent B16-F10 cells were harvested and resuspended in PBS at a concentration of 5.0 × 10^6^ cells/mL. the subcutaneous melanoma model was established by injecting 0.1 mL of the cell suspension into the right axilla of the male C57 BL/6N mice. Once the tumor volume was approximately 50 mm^3^ (5–7 days after implantation), the mice were randomly divided into four groups (*n* = 6 each) and treated accordingly: (ⅰ) the control group (PBS); CPPS-II (10 mg/kg); Dox (1 mg/kg); and DOX (1 mg/kg) + CPPS-II (10 mg/kg). DOX.HCl was administered via the tail vein and CPPS-II was administered via the intraperitoneal injection, respectively every 2 days for 8 days. All animals were handled in compliance with internal guidelines, and the above animal protocols and operations were approved by the ethics committee of the Chinese Academy of Medical Science, Beijing, China.

#### 4.5.2. Antitumor Efficiency

The antitumor effect was performed by the previously published work [39]. Briefly, the tumor volume and body weight were recorded at 2-day intervals. Tumor volume was monitored by measuring the perpendicular diameter with a caliper. The estimated tumor volume was calculated as 0.5 × length × width^2^. Tumor suppression rates (TSR) and tumor growth rates were calculated at the end of the study. Tumors and vital organs (heart, liver, spleen, lung, and kidney) were excised from the B16F10 tumor-bearing mice, and conduct the standard hematoxylin and eosin (H&E) staining test to evaluate the anti-tumor efficacy and systemic toxicity.

#### 4.5.3. Immunohistochemical Staining Analysis

In addition, immunohistochemical staining was performed on paraffin-embedded tumor sections using primary antibodies for Ki67 (dilution rate as 1:500, Abcam, Cambridge, UK), a cluster of differentiation 31 (CD31, dilution rate as 1:800, Abcam, Cambridge, UK), matrix metallopeptidase 9 (MMP-9, dilution rate as 1:500, Abcam, Cambridge, UK) with biotinylated goat anti-rabbit IgG (dilution rate as 1:200, Zhongshan Jinqiao Biotechnology, Beijing, China) as the secondary antibody. Representative images were obtained using an inverted microscope (Leica, Weztlar, Germany). The expression levels of antigens were semi-quantitatively analyzed at 20× magnification using Image-Pro Plus software (Media Cybernetics, Rockville, MD, USA).

#### 4.5.4. TUNEL Assay

To further characterize apoptosis levels of the tumor section, TUNEL assays were performed following the steps recommended by the manufacturer (Beyotime, Shanghai, China). Biotin-antibody diluent was dropped on the sections for 3′-hydroxyl termini of DNA double-strand breaks staining and incubated for 60 min at 37 °C. Next, streptavidin-HRP was used as the secondary antibody for 40 min, followed by DAB staining and hematoxylin counterstaining. Positive staining was analyzed under an inverted microscope (Leica, Germany) at 20× magnification and analyzed on Image J.

#### 4.5.5. Analysis of Macrophage Polarization by Flow Cytometry

Macrophages infiltrating the tumor section were evaluated by flow cytometry. tumor tissues were harvested, minced, and incubated with serum-free RPMI-1640 medium. After digested by motor grinding (GentleMACS™, Miltenyi Biotec, Bergisch Gladbach, Germany), The dissociated cells were passed through the 40-μm nylon mesh and lysis of the red blood cells (RBCs). To stain macrophage cluster, anti-CD45 FITC (isotype control: Mouse IgG1, κ), an-ti-F4/80 APC (isotype control: Rat IgG2b, κ), anti-CD11b PE (isotype control: Mouse IgG1, κ), anti-CD206 PerCP (isotype control: Rat IgG2a, κ), anti-IAIE Brilliant Violet 421 (isotype control: Rat IgG2b, κ) were added to approximately 2 × 10^6^ cells suspended in PBS with fluorescent probes and incubated for 30 min. All staining reactions were performed in a final volume of 100 μL at 4 °C. Data was acquired using an 18-color flow cytometer (LSRII, BD, USA) and analyzed using FlowJo v10.0 software (Tree Star Inc., Ashland, OR, USA).

#### 4.5.6. Immunofluorescence Examination of Macrophages

Paraffin-embedded tumor sections were dewaxed with a gradient ethanol solution, then boiled in citrate buffer for antigen repair. Next, the primary antibodies were added to the surface of the sample slide and incubated in a wet box at 4 °C overnight, followed by the secondary antibody incubation at room temperature for about 50 min in the dark. All the antibodies are diluted 200 times for use. DAPI was used to counterstain nuclei for 10 min. The primary antibodies included: rat polyclonal CD206 antibody (Abcam, Cambridge, UK) and rat polyclonal iNOS antibody (Abcam, Cambridge, UK). CD206 and iNOS were used to label anti-inflammatory (M2) and pro-inflammatory (M1) macrophages, respectively. Fluorescence images were obtained by using a Nikon confocal microscope (SONY, Tokyo, Japan), and the average fluorescence intensity was quantitated by ImageJ (NIH, Bethesda, MD, USA). All staining experiments were performed three times with different sections.

#### 4.5.7. ELISA Experiments

Two commercial ELISA kits (IL-10&TGF-β, Invitrogen, Waltham, MA, USA) were used to evaluate IL-10&TGF-β levels in tumor tissues. Assays were performed according to our previously reported work [41]. 100 mg tumor tissues were homogenized in 1 mL PBS and stored overnight at −20 °C, Next, the samples were subjected to two freeze–thaw cycles and centrifuged at 4 °C for 10 min to separate supernatants. the concentration of total protein in supernatants was analyzed by the BCA method (Beyotime, Shanghai, China). The sandwich ELISA method was performed according to the manufacturer’s instructions, and OD values were measured at 450 nm using a microplate reader. The standard curve was generated using a known concentration of recombinant cytokines and used to calculate the cytokine concentration of the samples.

#### 4.5.8. Western Blot Assay

The tumor tissue homogenate solution was lysed by lysis buffer containing protease inhibitors to extract the total proteins. Then, the samples were loaded onto 10% SDS-PAGE gels of separation and transferred to the PVDF membrane (Millipore Sigma, St. Louis, MO, USA). Bovine serum albumin (5%) in Tris-buffer saline containing 0.1% Tween-20 (TBST) was used to block the unspecific proteins for 1 h at room temperature. The primary antibodies were used by diluting 1000 times with TBST buffer. After incubating with the following primary antibodies: phosphorylated Stat1 (Abcam, Cambridge, UK), Stat1 (Abcam, Cambridge, UK), phosphorylated Stat3 (Abcam, Cambridge, UK), or Stat3 (Abcam, Cambridge, UK). Rabbit anti-β-actin (Perprotech, China) was employed as a loading control, the membranes were incubated with HRP-labeled secondary antibodies for 1 h at room temperature. The secondary antibody was used by diluting 2000 times with TBST buffer. Finally, protein expression was quantitatively measured and visualized by SuperLumia ECL HRP Substrate Kit (Servicebio) and Image J.

### 4.6. Statistical Analyses

Data were presented as mean ± standard deviation (SD) of a minimum of three replicates unless otherwise indicated. Comparisons among multiple groups were conducted by one-way ANOVA followed by analysis using GraphPad Prism 8.0 (La Jolla, CA, USA). *p* values < 0.05 and <0.01 were considered to indicate statistical significance.

## 5. Conclusions

We extracted three polysaccharides with different molecular weights from *Codonopsis pilosula* and compared their antitumor activities. According to our results, among the three polysaccharides, CPPS-II displayed the strongest antitumor effects, and only CPPS-II demonstrated satisfactory immunomodulatory activities. Second, CPPS-II showed potentiation and toxicity reduction effects on Adriamycin for cancer treatment, and it significantly improved cancer treatment efficiency and immune function. Therefore, CPPS-II is expected to be useful as an effective cancer treatment or adjuvant therapy, with important clinical implications for the chemotherapy of breast cancer.

## Figures and Tables

**Figure 1 pharmaceuticals-16-00895-f001:**
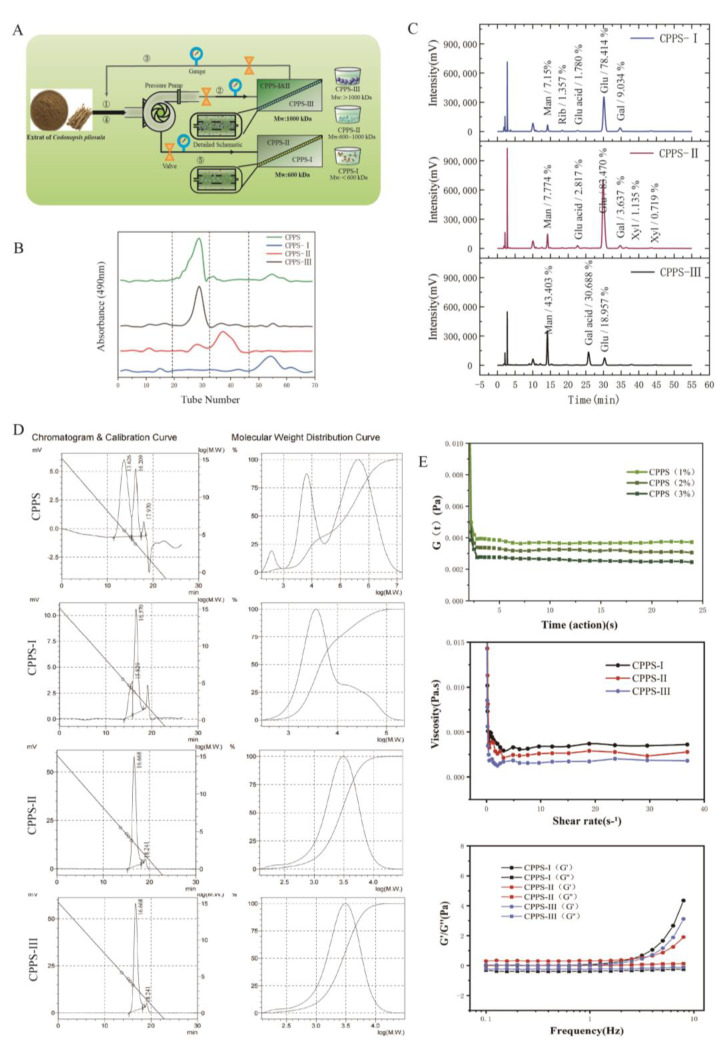
(**A**) Separation of crude polysaccharides by ultrafiltration membrane. (**B**) Elution curve (**C**) monosaccharide composition (**D**) and molecular weight of CPPS-I, II, and III purified by Sephadex G-100. (**E**) Rheological properties of polysaccharides.

**Figure 2 pharmaceuticals-16-00895-f002:**
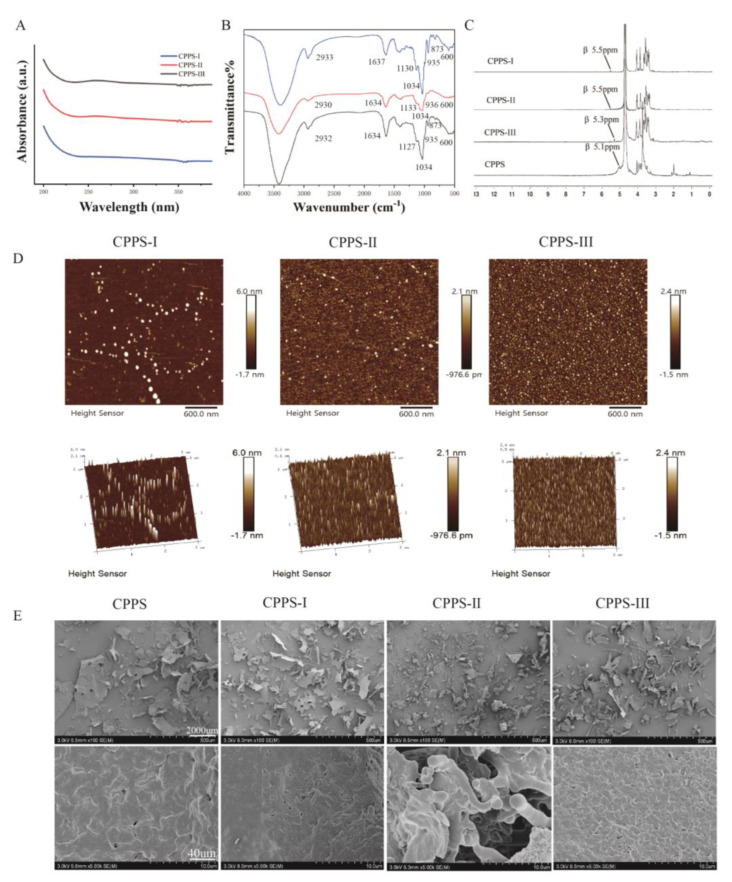
(**A**) UV, (**B**) Infrared, (**C**) NMR spectroscopy, (**D**) AFM, (**E**) SEM images of CPPS-I, CPPS-II, and CPPS-III.

**Figure 3 pharmaceuticals-16-00895-f003:**
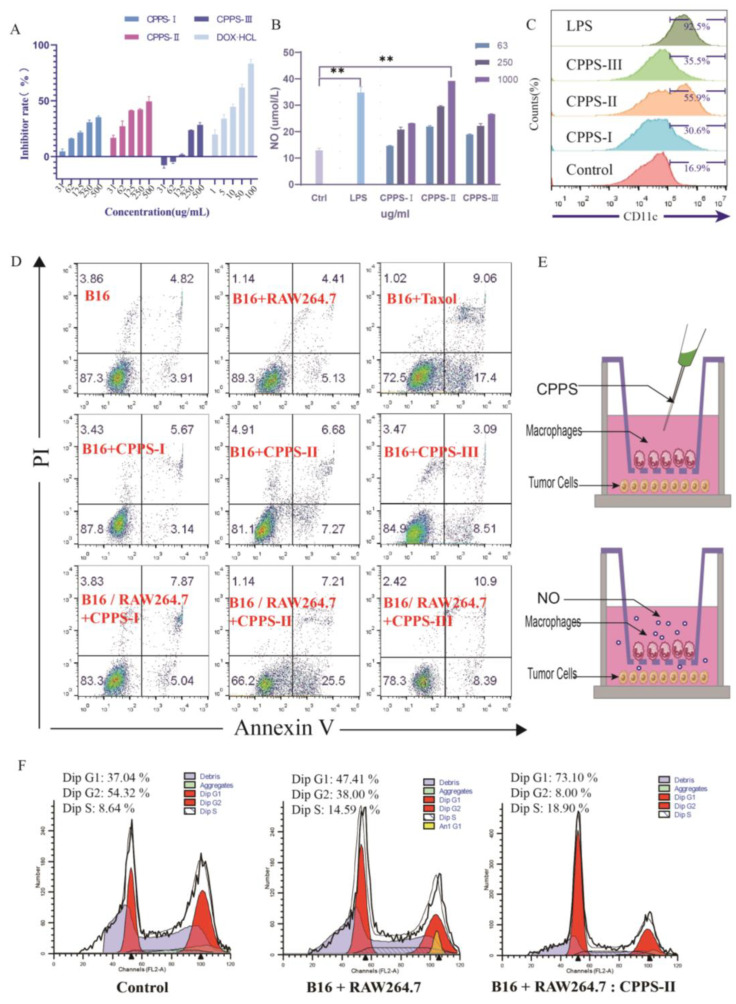
(**A**) Tumor inhibition rate of different *Codonopsis* polysaccharides and (**B**) NO production of every group by RAW264.7 cells. (**C**)Effects on macrophage polarization. (**D**) Flow cytometry analysis of apoptosis induced by CPPS in B16 cells using Annexin V-FITC staining. (**E**) Raw264.6 and B16 cells co-culture diagram. (**F**) Cell cycle distribution by flow cytometry of B16 cells.** *p* < 0.01.

**Figure 4 pharmaceuticals-16-00895-f004:**
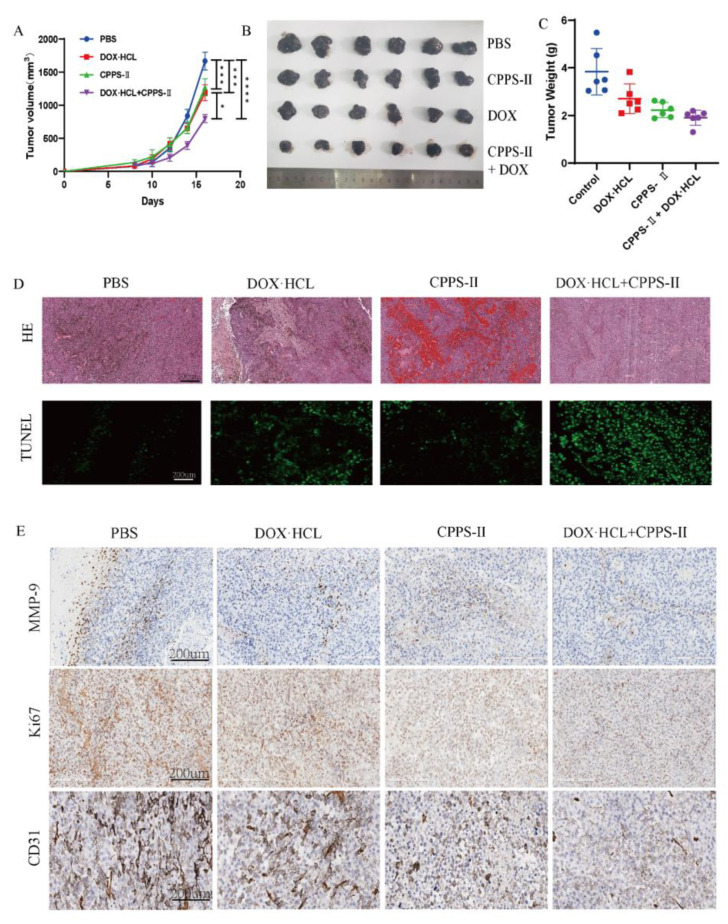
In vivo anti-tumor assay. The results of tumor growth curve (**A**), The typical photographs of the isolated tumor tissues (**B**), and tumor weight (**C**) of C57 mice bearing B16 tumor. * *p* < 0.05, *** *p* < 0.01, **** *p* < 0.001. (**D**) H&E staining, (**D**) TUNEL immunofluorescence, (**E**) MMP-9 staining, Ki67 staining, and CD31 staining of tumor tissues. Scale bar = 200 μm.

**Figure 5 pharmaceuticals-16-00895-f005:**
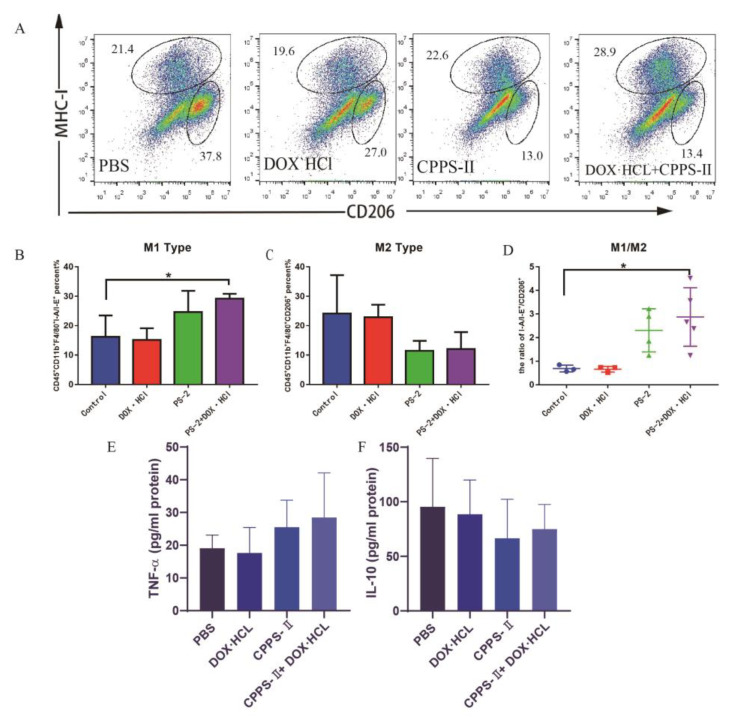
Exploration of anti-tumor immune mechanism. (**A**) the representative flow cytometry diagram, the statistical expression of M1 (**B**) and M2 (**C**) in tumor tissues, and the ratio of M1/M2 (**D**) were determined by FACS. (**E**,**F**) Cytokine expression in tumor tissues. * *p* < 0.05.

**Figure 6 pharmaceuticals-16-00895-f006:**
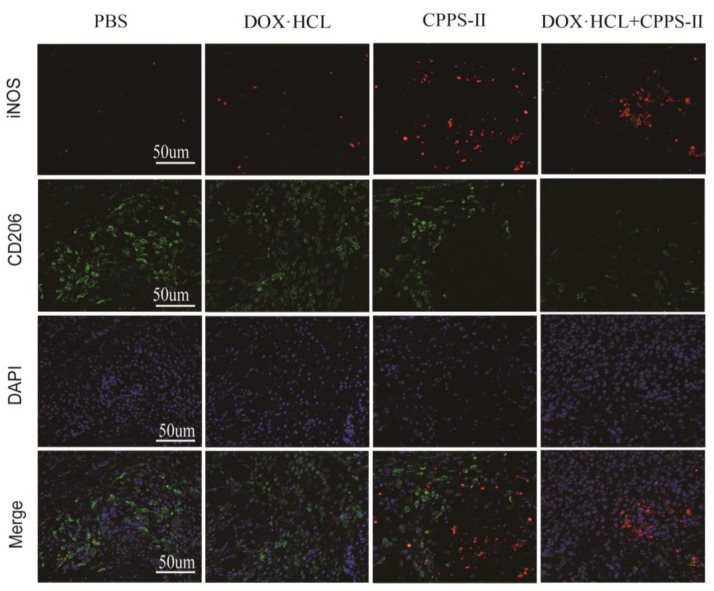
Exploration of anti-tumor immune mechanism. Immunofluorescence pictures of iNOS, CD206, and DAPI in tumor-bearing mice from each treatment group at the end of periods. Scale bar = 50 μm.

**Figure 7 pharmaceuticals-16-00895-f007:**
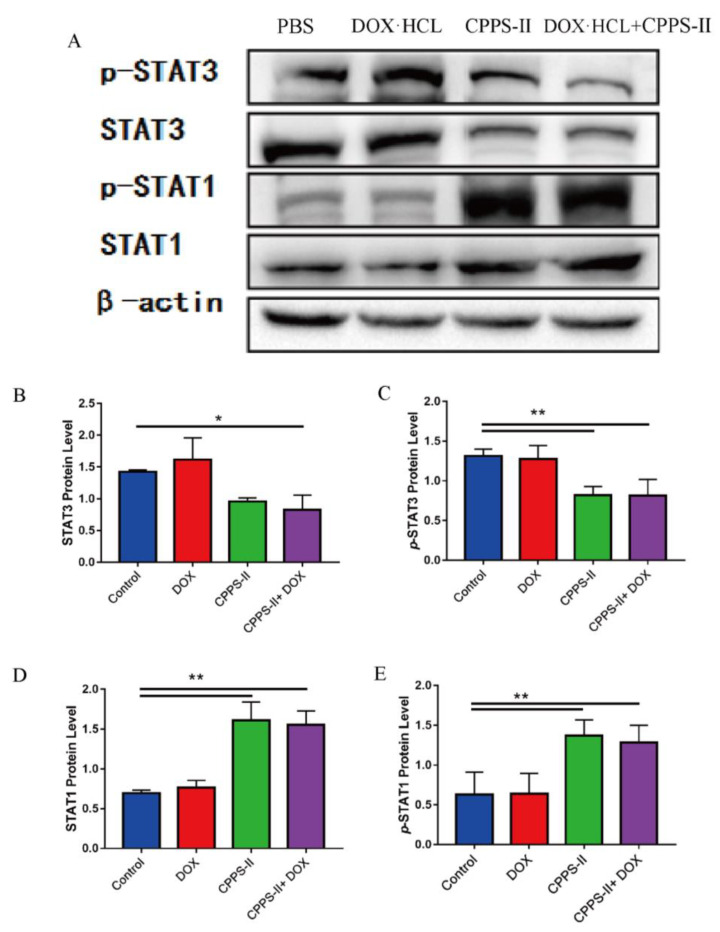
Signaling pathway determinations. (**A**) Western blot analysis of Stat3 (**B**) p-Stat3 (**C**) Stat1 (**D**) and p-Stat1 (**E**) expression in the B16 tumor after different treatments. * *p* < 0.05, ** *p* < 0.01.

**Figure 8 pharmaceuticals-16-00895-f008:**
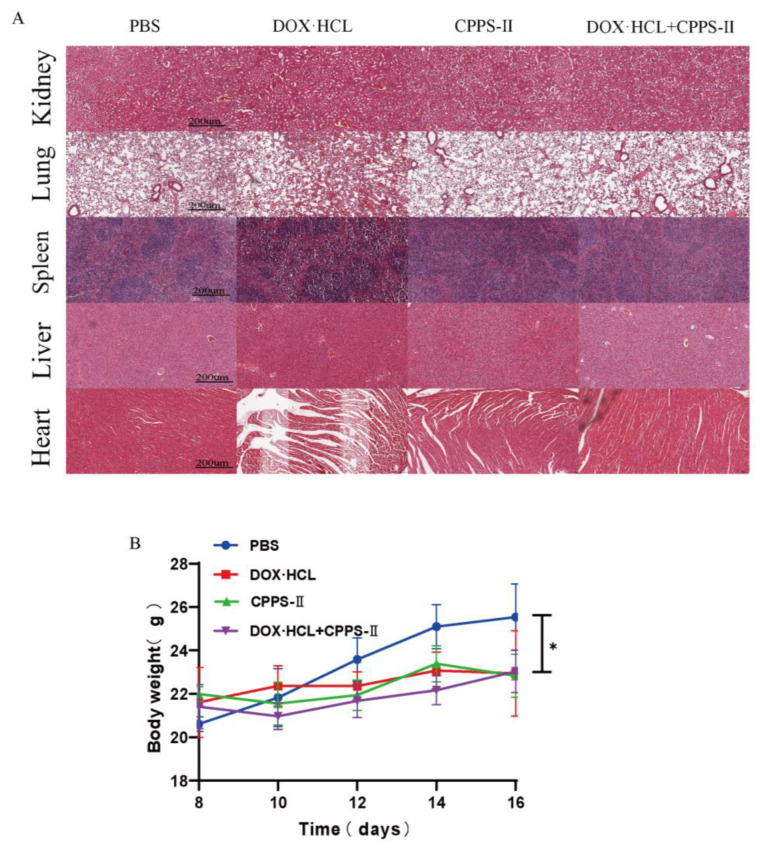
Safety testing of drugs. (**A**) H&E staining of heart, liver, spleen, lung, and kidney tissues of mice in each experimental group after 16 days of treatment. (**B**) The variation curves of body weight during the therapy period. * *p* < 0.05.

## Data Availability

Data is contained within the article.

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
