# Peer review of "Isolation, Purification, and Structural Characterization of Polysaccharides from Codonopsis pilosula and Their Anti-Tumor Bioactivity by Immunomodulation"

_pharmaceuticals, 2023, doi:10.3390/ph16060895_

Round 1

Reviewer 1 Report

The manuscript submitted by Nan Li et al, describes the Isolation, purification, and structural characterization of polysaccharides from Codonopsis pilosula. Moreover, the authors evaluate these polysaccharides on in vivo and in vivo models.

The manuscript would be interesting to publish but first major changes are needed.

In general, the figures are very small, and difficult to understand.

I find it quite surprising that polysaccharides can be analyzed in TEM (Figure 2E); in fact it is not clear the purpose of this experiment. Moreover, there seems to be a confusion between NMR, SEM and AFM (lines 484-489). These are completely different techniques and must be detailed in independent paragraphs.

Figure 3 B : there is an overexpression of Nitric Oxide in the samples treated with CPPS. How can the authors be sure that this does not affect the normal cells?

Did the authors evaluate the antiproliferative effect on normal cells?

Lines 273-274 : please, review the punctuation.  

Figure 6 : a strong magnification is desirable to be convincing on the colocalization.

Lines 304-324 : « Stat » et « stat », « STAT » : the authors must standardize.

Figure 7 A : please provide the whole Western gel image (in supplementary data).

Cell viability assay, NO measurement, Annexin V-FITC : please specify the exact name of the kit, the supplier and the reference.

Line 515 : « cell cycle was performed as previously described” : please provide the reference.

Line 525 “…and treated with CM” : please specify what is “ CM”.

Line 538: “were administered via the tail vein and the intraperitoneal injection respectively”: which way for which treatment?

Line 543; “The antitumor effect was performed by the previously published work” please give the reference.

Paragraph 4.5.3: the authors should specify the antibody concentrations used.

Paragraph 4.5.5. “Analysis of macrophage polarization by flow cytometry”: there is no isotype control.

Paragraph 4.5.6. “Immunofluorescence examination of macrophages” “Paraffin-embedded tumor section were dewaxed with a gradient ethanol solution, then boiled in citrate buffer for antigen repaired….”.  This protocol is not very precise, it must be detailed. The concentration of primary and secondary antibodies must be specified.

 Paragraph 4.5.7. « ELISA experiments » : the authors claim to have used three commercial ELISA kit, but they only put 2 (IL-10&TGF-β)

Lines 592-593 : « Assays were performed according our previously reported work”: please give the reference.

Paragraph 4.5.8. « Western blot assay » : The concentration of primary and secondary antibodies must be specified. What does ECL mean (line 611)?

Author Response

尊敬的教授:

根据意见和要求,我们对原稿进行了大量修改。在这里,我们附上了 word格式的修订稿,供您批准。还总结了一份回答裁判员每个问题的文件。附上一份带有黄色标记的更正部分的修订稿作为补充材料,以便于检查。我们修改了稿件中的相关部分。

Reviewer 2 Report

The manuscript Li et al describes the anti-tumor effect and immunomodulatory effect the Codonopsis pilosula root extract, more specific, the polysaccharides. To gain more insight in the relevant biologically-active polysaccharides, the authors prepared three groups based on different molecular weight ranges. These different fractions were analyzed with regard to certain chemical/structural parameters and the biological activity towards cancer cells and macrophage polarization. Although I am not a specialist in natural products or carbohydrates, this is an interesting manuscript to further elucidate natural compounds for cancer therapies.

However, the first in vitro assays are not the most convincing experiments. Some issues related to the viability assay and Annexin/PI assay are mentioned below. At least differences between the polysaccharide fractions could be determined. Very interesting was the in vivo part of this manuscript demonstrating a quite powerful product of the most active fraction.

There are some points and typing errors which need to be clarified/addressed as mentioned below:

L108-110: The description for Sephadex chromatography and GPC are somewhat confusing for the me. If I understand it correctly, both assays were used for checking the purity of the compounds and the Sephadex column was not used for an additional purification step as written in the experimental section. Why did the authors check the purity by two different GPC methods? In line 109 is a typing error: you mean G-100 instead of G-200?

L124-126: For the molecular weights it is better to write Mw for the average molecular weight and “polydispersity index” instead of dispersion coefficient.

L129: Polysaccharides are rich

L148 Figure 1 C and D the labeling of the graphs is poorly legible and the assignments in Fig 1D are missing. Please improve this!

L153: What type of purity is shown by the UV/Vis measurement? The absence of protein? This should be mentioned.

L165: Full-stop is missing after the [4].

L165ff: I am not sure if this statement is correct. I am not a carbohydrate specialist but normally, the alpha proton is more downfield shifted than the beta protons. Generally, I think for a fully determination of the chemical structure by NMR, other techniques need to be included and this is not necessary for this study.

L186 Figure 2: The labeling of the UV/Vis and IR graphic is hard to read. Similarly, the scale bars for the AFM picures.

L188: typing error “topography”

L192: The cell viability measurement shows an inhibiting rate. In the experimental section it is written that the B16 viability is normalized to treated Raw cells? What is the intention of this normalization? Please give the cell viability compared untreated cells! If the viability data for the Raw are available, they can be included as well to demonstrate preferential CPPS toxicity for tumor cells.

L195: highest concentration

L202: Why was the M1 polarization only identified by the NO release? Later the authors fully characterized M1 and M2 macrophages from tumor samples. Here I would like to see more data for the characterization. Is the CPPS-II potent enough to re-polarize M2 to M1 macrophages or is it only possible to polarize from M0 to M1?

L210ff Co-culture:

The authors said there was no change in apoptosis/necrosis rate after B16 tumor cells were co-cultured with macrophages. But why is the necrosis increasing from 3% to ~9%? Moreover, there must be some technical issues with this assay: The taxol treatment did not show any increase in apoptosis rate just an increase in necrosis. Normally, there is a general shift in both numbers. And why is the necrosis in the B16+CPPS-1 higher than for B16+Raw+CPPS-1? In the B16+Raw group is the necrosis similar to the B16+Raw+CPPS-II while in last group the apoptosis rate is increased. Generally, this experiment is not convincing at all but gives a trend.

My question to the authors: Have the authors carefully collected all (dead) cells in the upper medium prior to the sample preparation?

Similar is true for the cell cycle determinations, it seems that the Raw macrophages already had an M1 polarization shift, probably due to the cultivation. For M0 polarized macrophages no changes would be expected what the authors also mention. This I cannot see for none of the in vitro assays!

Most convincing and interesting are the in vivo experiments! In this section the anti-tumor activity and macrophage repolarization properties are clearly demonstrated!

L288 Figure 5: Increase the font size for the percentages in the FACS data!

L304 Signaling pathway

Please write exactly that tumor lysates were analyzed. It was unclear until reading the materials and methods section that tumor samples were used and not the cell lines.

L485: Wrong heading

L497: How many cells were exactly seeded and how much volume. I assume it was 100µl and 10,000 cells? This is an important information since the initial cell density has an effect on the treatment efficacy.

The same is true for the other in vitro assays! For Annexin and cell cycle assays, the abbreviation CM relates to conditioned media? Was only conditioned media used or the co-culture model with inserts? How many Raw cells were seed in the inserts?

L527 typing error: 5 lL = 5µL?

Flow cytometry and IHC: Add the antibody dilution for all antibodies.

Author Response

Response to Reviewer 2Comments

Dear professor:

Based on the comment and request, we have made extensive modification on the original manuscript. Here, we attached revised manuscript in the formats of  word for your approval, please see the attachment.. A document answering every question from the referees was also summarized. A revised manuscript with the correction sections yellow marked was attached as the supplemental material and for easy check purpose.We amended the relevant part in manuscript.

Round 2

Reviewer 1 Report

“Point 7: Figure 7 A : please provide the whole Western gel image (in supplementary data).”

The authors do not provide the original Western gel image but only modified images.

Author Response

Dear Reviewer:

I am glad to receive your acceptance email. I have uploaded the original western data in the minor revision via email, but due to the system, the reviewer did not see the data, so I have made the comment again. I am a PhD student who is about to graduate and I am very anxious to publish this article.Thank you for your understanding and kind help, and I am looking forward to hearing from you!

Kind regards,

Your Sincerely, Nan Li
